# Gradual Temperature Rise in Radiofrequency Ablation: Enhancing Lesion Quality and Safety in Porcine Myocardial Tissue

**DOI:** 10.3390/bioengineering12040360

**Published:** 2025-03-31

**Authors:** Cheol-Min Lee, Jae-Young Seo, Jin-Chang Kim, Min-Ku Chon

**Affiliations:** 1Department of R&D Center, TAU MEDICL Inc., Busan 50612, Republic of Korea; 2Department of Internal Medicine, School of Medicine, Pusan National University, Busan 46241, Republic of Korea; shope2024@pusan.ac.kr; 3Department of Internal Medicine, Pusan National University School of Medicine and Cardiology, Cardiovascular Center and Research Institute for Convergence of Biomedical Science and Technology, Pusan National University Yangsan Hospital, Yangsan 50612, Republic of Korea

**Keywords:** radiofrequency ablation, temperature-controlled, gradual temperature rise, turn-up time

## Abstract

Radiofrequency ablation (RFA) is a pivotal therapeutic technique for various medical conditions, including cardiovascular disease and oncological conditions such as liver and lung cancer. The energy-controlled mode in RFA procedures allows for uniform energy delivery but is less safe compared to the temperature-controlled mode. Therefore, it is necessary to develop a protocol that ensures safety while efficiently delivering energy in the temperature-controlled mode. In this study, we compared lesion formation using the gradual-temperature-rise mode to the fixed-temperature mode. We evaluated the lesion size, energy, cumulative time efficiency, and procedural safety in both in vitro and in vivo experiments with porcine myocardial tissue. Three experimental groups (n = 6) were compared to assess the effect of gradual-temperature-rise and fixed-temperature ablation modes. Five experimental groups (n = 6) were used to determine the optimal temperature turn-up time. The gradual-temperature-rise mode ablated larger lesions (10.48 ± 0.56 mm) compared to the 75 °C (7.67 ± 0.37 mm) and 85 °C (8.05 ± 0.36 mm) fixed-temperature groups (*p* = 0.002). The optimal turn-up time for efficient lesion formation was found to be between 120 and 180 s. The in vivo experiments validated the safety and efficacy of the optimized gradual-temperature-rise mode. Therefore, using the gradual-temperature-rise mode of temperature-controlled RFA enhances lesion formation, energy transfer, and safety, making it a promising approach for clinical application in cardiac ablation procedures.

## 1. Introduction

Radiofrequency ablation (RFA) is a therapeutic technique that is widely used in the treatment of various conditions, including cardiovascular disease and oncological conditions such as liver and lung cancer [1,2,3]. The RFA is a minimally invasive procedure that generates localized thermal energy to induce tissue necrosis, thereby enabling precise tissue ablation. In oncology, RFA is particularly useful for the treatment of early-stage hepatocellular carcinoma [4] and localized lung neoplasms [5]. This offers a nonsurgical alternative for patients ineligible for surgery because of comorbidities or tumor location. RFA’s success depends on its ability to enhance clinical outcomes by selectively ablating abnormal tissue while minimizing damage to the surrounding healthy structures [6].

Beyond oncology, RFA has become an essential therapeutic tool in cardiac interventions, playing a crucial role in managing arrhythmias [1,7] and structural heart conditions such as hypertrophic obstructive cardiomyopathy (HOCM) [8,9]. In these applications, radiofrequency energy is delivered via a catheter-based system to create thermal lesions for the ablation of pathological myocardial tissue. This minimally invasive method improves patient outcomes and quality of life by offering shorter recovery times and lower complication rates than surgical alternatives. The energy-controlled mode is the most commonly used ablation method in RFA procedures. This method delivers a constant energy value between the electrode and ground pad during monopolar ablation. Recent studies have highlighted the benefits and limitations of energy-controlled ablation [10]. While energy-controlled ablation is effective for creating lesions, it is less predictable in terms of temperature distribution, leading to variable lesion sizes and risks such as overheating.

For instance, studies comparing energy- and temperature-controlled ablation modes have demonstrated that temperature-controlled methods provide more consistent lesion formation and lower rates of complications such as tissue charring and steam pops [11,12]. The temperature-controlled mode is particularly advantageous for procedures requiring high precision, such as cardiac ablation, in which it is essential to minimize collateral damage to healthy tissue [13]. However, a limitation of temperature-controlled ablation is the need for precise target temperature monitoring and regulation, often requiring advanced equipment and operator expertise. Despite these challenges, temperature-controlled methods are increasingly being adopted as safer and more effective alternatives to energy-controlled ablation techniques [13,14,15,16,17]. However, temperature-controlled ablation has limitations due to its inherent power-adjustment mechanisms. As the RF generator dynamically increases and decreases the power levels to maintain a fixed electrode temperature [11,18], the total energy delivered to the target tissue can be lower than the target energy level [19], potentially reducing the efficiency of lesion formation. This can result in suboptimal outcomes, particularly in patients requiring deeper or larger lesions. Therefore, although temperature control offers safety benefits, careful consideration and protocol optimization are required to achieve sufficient lesion formation while maintaining efficacy.

In this study, we introduce a gradual-temperature-rise mode for RFA to overcome these issues. This approach improves energy transfer by reducing rapid surface heating, thereby allowing better energy penetration into deeper tissue layers. This gradual heating strategy improves lesion formation quality by promoting deeper and more uniform thermal penetration, reducing impedance spikes that limit energy transfer, and minimizing excessive carbonization at the tissue–electrode interface. Compared to the fixed-temperature mode, gradual temperature rise leads to more efficient energy injection patterns, optimizes tissue elasticity changes, and reduces the risk of steam pops or abrupt impedance changes, making it a safer and more effective approach for HCOM RFA.

## 2. Methods

### 2.1. RFA System and RF Catheter

The monopolar RFA catheters used in this study featured an outer diameter of 5 Fr, with an electrode length of 15 mm. The electrodes were fabricated from platinum–iridium (Pt/Ir) alloys in a flat-wire configuration, offering several advantages over traditional stainless-steel electrodes. The Pt/Ir electrodes exhibit superior electrical conductivity and radiopacity under X-ray imaging, ensuring enhanced procedural accuracy. Additionally, the soft material properties of Pt/Ir minimize mechanical stress on the tissues and vasculature, thereby reducing the risk of procedural complications. The coil structure of the electrodes provides high flexibility, making them well-suited for navigation through curved or tortuous vascular pathways. The flat-wire design, formed into a coil structure, ensures optimal wall contact, thereby enhancing energy transfer efficiency (Figure 1). The ablation catheter incorporates an internal cooling system for the electrodes to prevent the carbonization of coil electrodes, significantly improving the efficiency of RF energy delivery [18,20]. This cooling mechanism is critical for maximizing ablation efficiency in the temperature-controlled mode and ensuring consistent lesion formation without compromising tissue integrity. The ablation procedures were conducted using an RF generator (CAS20, RF Medical, Ltd., Seoul, Republic of Korea) operating at a high frequency of 480 kHz, equipped with a temperature-controlled mode. The RF generator operates at an input voltage of 220 V, with the power output dynamically adjusted based on temperature feedback. The output energy, measured in watts, ranges from 3 W to 10 W depending on the set temperature, tissue impedance, and the presence of a cooling system. The maximum output energy is limited to 20 W. We applied proportional–integral–derivative (PID) control to the RF generator to achieve a stable and large ablation zone. The PID controller enabled precise regulation of the target temperature, ensuring consistent energy delivery while mitigating risks such as tissue charring and overheating, which are commonly associated with energy-controlled ablation modes [14,21,22]. The temperature control system was specifically designed to optimize safety and efficacy during ablation by maintaining stable thermal conditions. To further enhance the procedure, a saline pump (SP-8800, AMPall Co., Ltd., Seoul, Republic of Korea) was used to inject normal saline at a flow rate of 1–3 cc/min, forming an integral part of the internal cooling system. This system effectively prevented excessive heating at the electrode–tissue interface, improved energy transfer efficiency, and minimized the risk of thermal injury to the surrounding tissues. A temperature sensor positioned between the electrode and the cooling lumen of the catheter provided real-time monitoring of tissue temperature, facilitating precise thermal feedback during ablation. Additionally, the system continuously measured tissue impedance between the electrode and grounding pad, enabling dynamic adjustments to the energy delivery parameters. This integrated approach ensured improved procedural control, optimized energy delivery, and minimized collateral tissue damage (Figure 2).

### 2.2. In Vitro Experiment

All in vitro tests were conducted using porcine myocardial tissue samples of similar size. Each tissue sample was immersed in a 10 L normal saline bath maintained at 36 °C to simulate physiological conditions. The distance between the ground pad and myocardial tissue was fixed at 30 cm to maintain uniform resistance throughout all experimental procedures. In the first set of experiments, lesion formation characteristics during RFA were evaluated by comparing the three experimental groups (n = 6 per group). Internal cooling was not applied to facilitate a clearer comparison between the experimental groups. Ablation was performed under the following conditions: (1) a fixed temperature of 75 °C for 10 min, (2) a fixed temperature of 85 °C for 10 min, and (3) a gradual-temperature-rise mode, wherein the temperature was incrementally increased from 65 °C to 85 °C at a rate of 5 °C every 2 min over a total duration of 10 min. The initial temperature was set to 65 °C to reach an overall ablation time of 10 min. Each experiment was repeated six times (n = 6). Outcome measures for intergroup comparisons included the maximum ablated diameter, energy consumption, and impedance values. The second experimental setup determined the optimal turn-up time for a gradual temperature increase during ablation. Here, we define turn-up time as the interval of time during which the temperature is increased. For example, if the turn-up time is 120 s, the target temperature is increased by 5 °C every 120 s. Internal catheter cooling was performed at a saline flow rate of 2 cc/min in all experimental groups. Ablation began at an initial temperature of 60 °C, with temperature increments of 5 °C every 2 min until 85 °C was reached. Ablation was terminated when impedance exceeded 250 Ω, and five turn-up times of 30, 90, 120, 150, and 180 s were tested (n = 6 per condition).

### 2.3. In Vivo Experiment

Animal experiments were conducted in compliance with the guidelines of the Institutional Animal Care and Use Committee of the Pusan National University Hospital. Ethical approval for this study was obtained prior to experimentation. Ten pigs (42.8 ± 4.7 kg) were used for the in vivo validation of in vitro findings and to assess lesion formation under physiological conditions. General anesthesia was induced by intramuscular administration of alfaxalone (5 mg/kg) and xylazine (2 mg/kg) and maintained using isoflurane (3%). Intravenous amiodarone (150 mg bolus followed by 1 mg/min infusion) was administered as needed to mitigate catheter-induced arrhythmias. Continuous electrocardiography (ECG) monitoring was performed during the procedure to detect and manage potential adverse cardiac events. Additional precautions were considered for the blood flow-induced heat sink effect during ablation. RF ablation was performed within the interventricular septum using two temperature protocols to evaluate the lesion size, procedural safety, and ablation effectiveness. In the fixed-temperature mode, ablation was conducted for a pre-determined duration at either 75 °C or 85 °C. In the gradual-temperature-rise mode, the temperature was incrementally increased from 60 °C to 85 °C in 5 °C increments. The ablation end temperature varied from 75 °C to 85 °C. The gradual-temperature-rise mode involved a turn-up time of 180 s. However, the turn-up time was reduced to 120 s in cases where ECG abnormalities were detected. Internal cooling was maintained with a saline flow rate of 1–3 cc/min during all procedures to prevent tissue overheating and ensure uniform lesion formation.

### 2.4. Statistical Analysis

A Shapiro–Wilk normality test was conducted for all data, indicating that the data were normally distributed. Data are summarized as mean ± standard deviation. The in vitro data of the experimental groups were compared using one-way analysis of variance (ANOVA), followed by Tukey’s post-hoc test for multiple comparisons. Welch’s ANOVA was also used where appropriate. For the in vivo data, an independent sample *t*-test was used to compare the gradual and fixed-temperature groups. All statistical analyses were performed using R (R Statistical Software for Windows, version 4.2.1, 2022, Foundation for Statistical Computing), along with the R package PCMCRplus. A *p*-value of <0.05 was considered to be statistically significant.

## 3. Results

### 3.1. Temperature Mode Comparison

The first set of in vitro experiments compared lesion formation using different temperature control modes: fixed-temperature settings of 75 °C and 85 °C and a gradual temperature rise from 65 °C to 85 °C. The results indicated that the gradual-temperature-rise group achieved larger lesion sizes (10.48 ± 0.56 mm, n = 6) compared to the fixed-temperature groups at 75 °C (7.67 ± 0.37 mm, n = 6) and 85 °C (8.05 ± 0.36 mm, n = 6). These differences were statistically significant (*p* < 0.001) (Figure 3 and Table 1). In addition to lesion size, lesion uniformity was assessed based on lesion shape consistency and boundary definition. The gradual-temperature-rise mode produced lesions with well-defined circular geometries and smoother transitions between the ablated and unaffected tissue, whereas the fixed-temperature modes often resulted in more irregular lesion shapes with uneven thermal boundaries. This observation suggests that gradual heating allows for a more controlled and homogeneous energy distribution, minimizing excessive thermal gradients that can contribute to non-uniform lesion formation. Histological cross-sections further confirmed that lesions produced in the gradual-temperature-rise mode exhibited more consistent thermal penetration and reduced boundary irregularities, whereas abrupt heating in the fixed-temperature modes led to uneven lesion characteristics. In the gradual-temperature-rise mode, RF energy increases in regular 2 min intervals, corresponding to each temperature increment phase, resulting in a periodic rise in energy delivery. In contrast, the fixed 85 °C mode exhibited a rapid initial increase in energy followed by an early plateau, which was associated with a significant rise in impedance. This impedance increase, caused by early dehydration and tissue carbonization, restricted further energy transfer, leading to limited lesion growth despite the high target temperature. These findings suggest that a gradual temperature increase allows better energy penetration and distribution, reducing the risk of surface overheating and improving overall lesion uniformity.

### 3.2. Turn-Up Time Optimization

A second in vitro experiment was conducted to determine the optimal turn-up time for temperature increments during RFA. Five different turn-up times (30, 90, 120, 150, and 180 s) were tested. The maximum ablated myocardial diameter (17.42 ± 1.27 mm) was observed at a turn-up time of 180 s. The lesion area consistently increased with longer turn-up times, with the largest lesion area (318.76 ± 41.57 mm^2^) recorded at 180 s. A turn-up time of 120 s provided the best balance between lesion formation and ablation efficiency, producing a lesion area of 263.81 ± 28.49 mm^2^ with a cumulative ablation time of 769.5 ± 103.12 s. Additionally, the energy consumption at 120 s (7939.5 J) was moderate compared to the higher consumption observed with extended turn-up times. Shorter turn-up times, such as 30 and 90 s, resulted in smaller lesion areas with reduced energy requirements but were less effective in achieving larger and clinically relevant lesion sizes. The impedance profile analysis further supported these findings. Shorter turn-up times led to abrupt impedance changes, which disrupted stable energy delivery and resulted in non-uniform lesion formation. Longer turn-up times, particularly at 120–180 s, facilitated a gradual impedance increase, maintaining stable energy transfer and allowing deeper thermal penetration. The energy consumption increased progressively with longer turn-up times, reaching its highest value at 180 s (10,688.67 J). Therefore, a turn-up time of 120–180 s provided the most effective balance between lesion size, uniformity, energy efficiency, and impedance stability, optimizing both procedural efficacy and ablation performance relative to the cauterization time (Figure 4 and Table 2).

### 3.3. In Vivo Experimental Outcomes

In vivo experiments were conducted using a porcine model to validate the findings of the in vitro experiments under physiological conditions. Although the protocols optimized in vitro were applied as closely as possible, several variations were encountered. The blood flow-induced heat sink effect surrounding the catheter necessitated higher energy delivery compared to in vitro experiments. ECG changes were observed when temperatures exceeded 80 °C. A cooling saline flow rate of 1–3 cc/min was maintained throughout the experiments to prevent tissue overheating and ensure consistent lesion formation. Higher flow rates (2–3 cc/min) were particularly effective in stabilizing the electrode temperature and preventing surface overheating, thereby contributing to procedure safety. However, a flow rate of 2 cc/min was found to be optimal as a rate of 3 cc/min posed technical challenges due to the inner diameter of the cooling lumen being 0.016 in. To ensure fair comparisons between treatment conditions, the fixed-temperature mode was conducted at either 75 °C or 85 °C for a predefined duration, maintaining a stable temperature throughout the procedure. In contrast, the gradual-temperature-rise mode started at 60 °C and increased in 5 °C increments until the target temperature was reached, either 75 °C or 85 °C. The standard turn-up time was 180 s, but in cases where ECG abnormalities were detected, it was reduced to 120 s for safety reasons. This approach ensured that variations in thermal conditions did not affect the comparability of results between groups. RFA was performed on the interventricular septum of 10 pigs, with four undergoing ablations using the fixed-temperature mode (75 °C and 85 °C) and six using the gradual-temperature-rise mode. All animals were harvested immediately after tissue analysis. The results demonstrated that the maximum lesion diameter in pigs treated with the gradual-temperature-rise mode (13.83 ± 0.75 mm, n = 6) was significantly larger than in those treated with the fixed-temperature mode (9.25 ± 2.75 mm, n = 4; *p* = 0.004). To evaluate lesion uniformity, we assessed lesion shape consistency and boundary definition across different samples. The gradual-temperature-rise mode produced lesions with well-defined and more predictable geometries, whereas the fixed-temperature modes often resulted in irregular lesion shapes due to abrupt heating and uneven energy distribution. Histological cross-sections further confirmed this observation, revealing more homogeneous thermal penetration and fewer disrupted tissue boundaries in the gradual-temperature-rise mode. These findings indicate that the gradual-temperature-rise mode enhances lesion uniformity, leading to more consistent and effective ablation outcomes. Continuous ECG monitoring revealed mild non-fatal electrocardiographic changes at temperatures exceeding 80 °C in some pigs. No adverse cardiac events were recorded during the experiments (Figure 5). Pathological evaluation of the ablated tissues confirmed effective lesion formation with minimal collateral damage, highlighting the safety and efficiency of the gradual-temperature-rise mode in the porcine model. This protocol enabled efficient energy transfer, minimized the risk of tissue carbonization or thermal damage, and produced consistent lesion formation. Histopathological analysis of the ablated myocardial tissue was performed using hematoxylin and eosin (H&E) staining, which revealed features indicative of successful ablation. Macroscopically, the anterior interventricular septum exhibited a circular brown lesion with an approximate area of 143 mm^2^, and cardiomyocytes in the central region of the lesion displayed coagulative necrosis, with dilated blood vessels and an edematous stroma (Figure 6). In the peripheral lesion zone, myocytes exhibited eosinophilia, suggesting that the tissue damage extended beyond the central necrotic area. These observations are consistent with the characteristic thermal effects expected from the gradual-temperature-rise mode and confirm the efficacy of the protocol in achieving effective lesion formation.

## 4. Discussion

This study demonstrates the significant advantages of employing a gradual-temperature-rise mode of RFA over traditional fixed-temperature modes. By evaluating lesion formation, energy efficiency, and procedural safety through rigorous in vitro and in vivo experiments, we provide a comprehensive understanding of the benefits of gradual temperature modulation (Table 3). Our findings support the clinical potential of this approach, particularly for cardiac ablation procedures such as septal reduction in HOCM. The gradual-temperature-rise mode showed superior performance by creating larger and more uniform lesions compared to fixed-temperature settings of 75 °C and 85 °C. This enhanced efficacy is attributed to the controlled energy delivery facilitated by gradual heating, which allows deeper thermal penetration into the target tissues while minimizing surface overheating. The biophysical mechanism underlying this improvement involves stepwise energy delivery, which prevents early tissue dehydration and impedance spikes that typically limit energy transfer in fixed-temperature modes. Gradual heating reduces abrupt impedance increases by allowing progressive tissue conductivity changes, ensuring stable energy application throughout the procedure. Additionally, by maintaining a lower initial temperature and incrementally increasing it, the gradual-temperature-rise mode reduces carbonization and steam pop formation, which are common complications in conventional RFA methods. This mode avoids abrupt energy spikes and distributes heat more evenly, thus reducing the likelihood of tissue carbonization and steam pops, which are common complications in fixed-energy or temperature-controlled RFA modes [11,12]. These benefits are particularly important for delicate cardiac procedures wherein uniform lesion formation is critical for the prevention of arrhythmias or procedural failure [23,24]. The ability to ablate target tissues effectively without damaging surrounding structures marks a significant advancement in RFA technology and addresses the limitations associated with traditional fixed modes, particularly in cases of challenging anatomical regions [25,26,27]. A key finding of this study was the identification of the optimal turn-up time for temperature increments. The in vitro experiments revealed that a turn-up time of 120 s optimizes lesion size, energy consumption, and cumulative ablation time, whereas a 180 s turn-up time produces the largest lesion area and diameter. However, longer procedural duration and higher energy consumption are not practical in clinical settings, particularly when patient safety and time efficiency are of paramount importance [28,29,30]. Shorter turn-up times such as 30 and 90 s produced smaller lesions, highlighting the importance of gradual heating for effective lesion formation. Thus, the 120 s turn-up time demonstrated sufficient lesion formation with moderate energy consumption and manageable impedance levels. This finding is critical for developing customizable standardized protocols that align with current clinical guidelines for cardiac ablation procedures. The ability to control energy delivery while maintaining stable impedance and preventing excessive thermal damage enhances procedural safety, a key factor emphasized in guidelines for septal reduction therapy in HOCM patients. The gradual-temperature-rise mode allows for tailored ablation protocols based on lesion size requirements and patient-specific anatomical considerations, offering an adaptable approach to improving clinical outcomes. For more complex cases requiring larger lesions, extending the turn-up time to 180 s may be advantageous, provided that procedural constraints are considered. In vivo experiments using a porcine model validated the in vitro findings under physiological conditions, facilitating comprehension of the practical application of the gradual-temperature-rise mode. The gradual mode produced significantly larger lesions than the fixed-temperature mode, with no adverse events observed during the procedures. Continuous ECG monitoring indicated mild non-fatal ECG changes at temperatures over 80 °C, suggesting that the gradual-temperature-rise mode is safe even under the dynamic conditions of a living system. The blood flow-induced heat sink effect surrounding the catheter was mitigated by maintaining an optimal cooling flow rate of 2 cc/min, which stabilized the electrode temperature and prevented overheating. Although higher flow rates (3 cc/min) were effective for temperature control, they posed technical challenges owing to the dimensions of the cooling lumen. Therefore, the optimization of the cooling mechanisms is instrumental to balancing thermal management with procedural practicality. The pathological evaluation of the ablated myocardial tissues provided further evidence of the efficacy and safety of the gradual-temperature-rise mode. H&E staining revealed classic histopathological features of successful ablation, including coagulative necrosis in the lesion center, dilated blood vessels, and edematous stroma. The peripheral lesion region exhibited eosinophilic changes, indicating effective thermal damage extending beyond the lesion core. Macroscopically, the ablated region presented as a well-demarcated circular lesion, consistent with the desired outcome of controlled ablation. These findings confirm that this protocol may be used to achieve precise and predictable lesion formation while minimizing collateral tissue damage [31,32]. Our findings have profound clinical implications, particularly for cardiac ablation procedures such as HOCM treatment. The gradual-temperature-rise mode offers a safer and more effective alternative to the fixed-temperature mode by providing consistent lesion formation and reducing procedural complications. Effective septal ablation is crucial for reducing left ventricular outflow tract obstruction, alleviating symptoms such as dyspnea and syncope, and improving hemodynamic performance in HOCM treatment [9]. The gradual-temperature-rise approach minimizes the risks of incomplete ablation or thermal injury, making it an attractive option for clinicians. Despite these promising results, this study has certain limitations that must be addressed in future research. While the porcine model provides valuable preclinical validation, it does not fully replicate human anatomy and physiological variations. Additionally, the sample size was relatively small, which may limit the generalizability of the findings. Future studies should focus on conducting clinical trials with larger and more diverse patient populations to confirm the efficacy and safety of this protocol in human subjects. Furthermore, investigating the impact of anatomical variability, disease severity, and comorbidities on lesion formation will be crucial in refining this technique for broader clinical applications. Advances in catheter technology and cooling system optimization could further enhance the effectiveness and practicality of the gradual-temperature-rise approach, paving the way for its integration into routine clinical practice.

## 5. Conclusions

A gradual-temperature-rise RFA mode is more effective in creating larger and more uniform lesions than a fixed-temperature mode. An optimal turn-up time of 120–180 s achieves a balance between lesion size, energy efficiency, and safety. Gradual heating enhances heat distribution, reduces complications such as steam pops, and improves energy delivery. Our findings suggest that the gradual-temperature-rise mode, when combined with an optimized protocol, can enhance RFA efficacy, particularly for HOCM treatment. Future studies are required to validate these results in clinical settings.

## Figures and Tables

**Figure 1 bioengineering-12-00360-f001:**
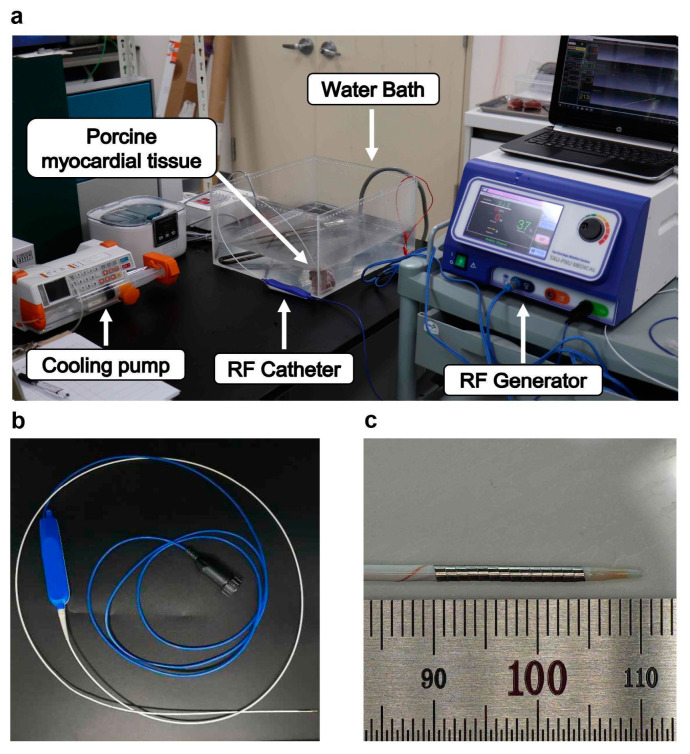
Experimental setup for in vitro ablation and RF catheter design. (**a**) In vitro ablation setup with porcine myocardial tissue. (**b**) RF catheter used in the experiment. (**c**) Detailed view of the RF catheter tip.

**Figure 2 bioengineering-12-00360-f002:**
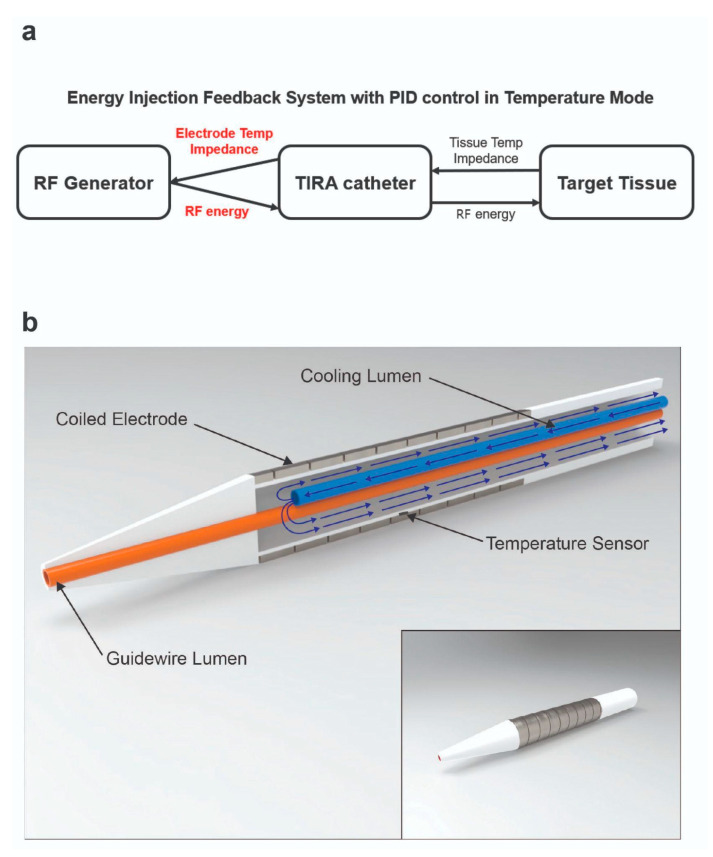
Special features of the RFA system. (**a**) RF generator feedback system with PID control. (**b**) Internal cooling system structure of the RF catheter.

**Figure 3 bioengineering-12-00360-f003:**
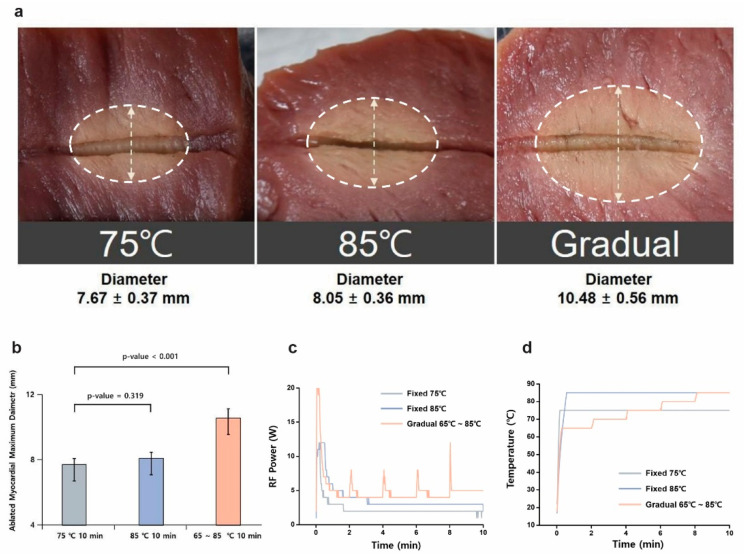
Comparison of lesion formation between fixed and gradual temperature modes. (**a**) Ablation lesions at fixed temperatures of 75 °C and 85 °C, compared to a gradual temperature rise (65–85 °C). (**b**) Maximum diameter of ablated myocardial tissues for each temperature mode (*p* < 0.001 between gradual and fixed-temperature modes). (**c**) Energy injection patterns in the gradual-temperature-rise mode and fixed-temperature modes. (**d**) Temperature variation over time for the fixed-temperature-control modes and gradual-temperature-rise mode (turn-up time of 120 s).

**Figure 4 bioengineering-12-00360-f004:**
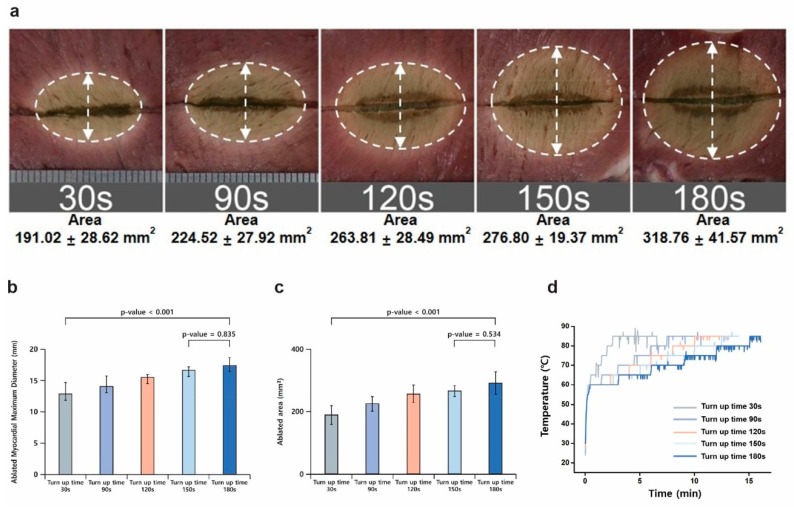
Effect of turn-up time on the formation of ablation lesions. (**a**) Ablation lesions for each turn-up time. (**b**) Maximum lesion diameter for each turn-up time. (**c**) Ablated area for each turn-up time. (**d**) Temperature changes according to turn-up time.

**Figure 5 bioengineering-12-00360-f005:**
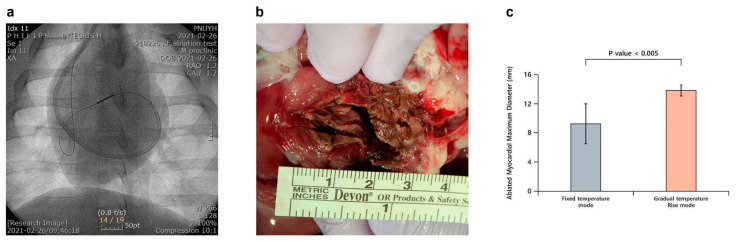
Results of in vivo experiments. (**a**) Angiographic view of the RFA catheter placed in the interventricular septum. (**b**) Gradual-temperature-rise mode (60–80 °C) for 15 min of ablation followed by harvest to observe the interventricular septum. (**c**) Comparison of maximum lesion diameter between fixed and gradual-temperature-rise modes (*p* < 0.005).

**Figure 6 bioengineering-12-00360-f006:**
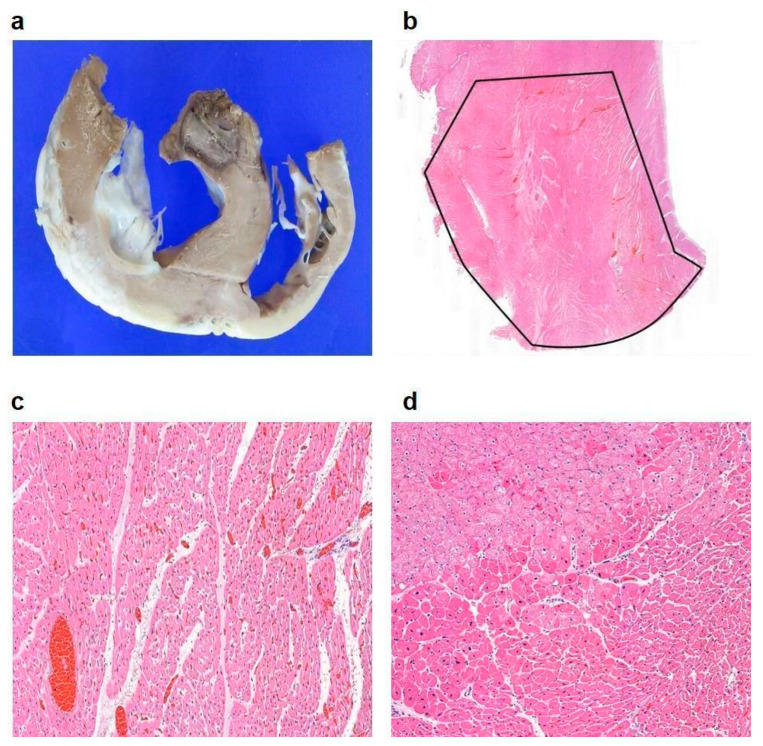
Histopathological analysis of in vivo experiments. (**a**) Gross specimen of the anterior interventricular septum showing a circular brownish lesion resulting from ablation. (**b**) The ablated lesion measures approximately 143 mm^2^ in area. (**c**) In the central portion, myocytes undergo coagulation necrosis. The vessels are dilated, and the stroma is edematous. (**d**) The peripheral myocytes in the lesion reveal eosinophilia.

**Table 1 bioengineering-12-00360-t001:** Results of the ex vivo experiments for the three considered groups (n = 6, mean ± standard deviation).

Group	Average Impedance (Ω)	Total Energy (J)	Ablated Myocardial Maximum Diameter (mm)
75 °C; 10 min	69.41 ± 8.28	1986.17 ± 334.24	7.67 ± 0.37
85 °C; 10 min	73.09 ± 11.88	2290.33 ± 355.60	8.05 ± 0.36
Gradual; 10 min	72.47 ± 4.48	2823.23 ± 489.17	10.48 ± 0.56

**Table 2 bioengineering-12-00360-t002:** Results of the ex vivo experiments according to the turn-up time (n = 6, mean ± standard deviation).

Turn-Up Time (s)	Average Impedance (Ω)	Total Energy (J)	Cumulative Time (s)	Ablated Maximum Diameter (mm^2^)	Ablated Area (mm^2^)
180	57.17	10,688.67	993.17 ± 44.14	17.42 ± 1.27	318.76 ± 41.57
150	55.6	8509.33	837.5 ± 36.36	16.65 ± 0.54	276.80 ± 19.37
120	61.97	7939.5	769.5 ± 103.12	15.52 ± 0.43	263.81 ± 28.49
90	62.72	5825.33	645.67 ± 148.09	14.07 ± 1.66	224.52 ± 27.92
30	64.07	4034.33	326 ± 115.89	12.87 ± 1.83	191.02 ± 28.62

**Table 3 bioengineering-12-00360-t003:** Results of the in vivo experiments in the porcine myocardium tissue.

Group	Cumulative Time (min)	Total Energy (J)	Average Impedance (Ω)	Maximum Diameter (mm)	*p*-Value
Fixed-temperature mode (n = 4)	13.25	7624.75	47.5	9.25 ± 2.75	0.004
Gradual-temperature-rise mode (n = 6)	14.17	9933.67	50	13.83 ± 0.75	0.004

## Data Availability

The original contributions presented in this study are included in the article. Further inquiries can be directed to the corresponding author.

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
