# Peer review of "Gradual Temperature Rise in Radiofrequency Ablation: Enhancing Lesion Quality and Safety in Porcine Myocardial Tissue"

_bioengineering, 2025, doi:10.3390/bioengineering12040360_

Round 1

Reviewer 1 Report

Comments and Suggestions for Authors

The article by Cheol-Min Lee et al. describes a set of comparisons between the gradual temperature rise mode vs temperature-controlled mode of Radiofrequency ablation (RFA) on in-vitro and in-vivo experiments with porcine myocardial tissue.

The reviewer greatly appreciates the efforts of the authors, especially in the quality of study design, experimental execution, writing and presentation. The reviewer has a RITA Radiofrequency Ablation System (AngioDynamics) in the lab and shares the frustration of controlling the ablative process. The temperature rise mode proposed by the authors has the potential to address some shortcomings of current RFA procedures.

Before endorsing for publications, there are a few concerns over the current version of the articles. To be specific:

(1) The implementation of “Gradual temperature rise in radiofrequency ablation”

Since this is a new control method, at least the reviewer hasn’t been aware of, more details of this new approach during RFA are needed. Even though some information, such as the diagram in Figure 2a, the overall descriptions are still oversimplified. Technical aspects, including but not limited to the following, are encouraged.

  • temperature sensor

Where exactly is the temperature and what temperature does it really measure? Figure 2b looks like it’s measuring the temperature of the “cooling lumen” other than the tissue in contact.

  • control/adjustment logic

How does the voltage, current or power of the RFA being controlled with temperature inputs?

  • temperature history during RFA

Examples of the temperature history would be helpful, especially demonstrating how well the system achieves “Gradual temperature rise”.

  • the turn-up time

Examples of the temperature history would help.

(2) The actual device(s)

It looks to me that a few devices were made (in section 2.1. RFA System and RF Catheter) and used in the experiments with different geometry: an outer diameter of 4–5 Fr: 1.33mm (for 4 Fr) to 1.67mm (for 5 Fr) and electrode lengths of 10–15 mm.

A fair comparison should be made using the same device, otherwise the difference in lesion formation can be attributed to different sizes not the modes of operation (constant temp. vs gradual temperature rise). Please confirm your device geometries of all the treatments.

(3) The overall advantages of the “gradual temperature rise mode” over “temperature-controlled mode” should be made clearer.

If one only considers size (the larger the better), the microwave ablation is the clear winner.

If the shape (the more spherical the better) is concerned, then a smaller probe of the RFA will be needed.

If the aim is to avoid carbonation (scar tissue) and vaporization (phase change), then lower temperature and longer RFA time would help.

In other words, what new capability does the “gradual temperature rise mode” offer that cannot be achieved by other means? What specific problems do the “gradual temperature rise mode” solve that cannot be solved by other designs?

Other minor questions or suggestions, in no particular order.

(1) The introduction has only 1 paragraph and is fairly long. Breaking it down to a few smaller ones is recommended.

(2) More clear definitions of the lesion quality (lesion formation quality) and safety is greatly appreciated, especially in the context of hypertrophic obstructive cardiomyopathy (HOCM) Radiofrequency ablation (RFA).

(3) How is the lesion defined? From figure 3a, a transient zone exists between the pink outer area and the inner white area.

(4) The first in vitro experiment, the reviewer does not understand why (ii) has smaller ablative size and lower energy than (iii). Wouldn’t maintaining the RFA probe at higher temperature (85C) require more power than that at a lower temperature (65-85)?

(i) a fixed temperature of 75°C for 10 min,

(ii) fixed temperature of 85°C for 10 min, and

(iii) gradual temperature rise mode, wherein the temperature was incrementally increased from 65°C to 85°C at a rate of 5°C every 2 min over a total duration of 10 min.

(5) The 2nd in vitro experiment, the following description on “Turn-up time” is confusing. Again, plots of temperature histories would help.

“The second experimental setup determined the optimal turn-up time for a gradual temperature increase during ablation. Turn-up time refers to the time interval in which the temperature increases by 5°C. Internal catheter cooling was performed at a saline flow rate of 2 cc/min in all experimental groups. Ablation began at an initial temperature of 60°C, with temperature increments of 5°C every 2 min until 85°C was reached. Ablation was terminated when impedance exceeded 250 Ω, and five turn-up times of 30, 90, 120, 150, and 180 s were tested (n = 6 per condition).”

(6) The in vivo experiment, what exactly are each of the treatment conditions? There are variations of the thermal conditions, making comparisons somewhat unfair.

“In the fixed temperature mode, ablation was conducted for a pre-determined duration at either 75°C or 85°C. In the gradual temperature rise mode, the temperature was incrementally increased from 60°C to 85°C in 5°C increments.  The ablation and temperature varied from 75°C to 85°C. The gradual temperature rise mode involved a turn-up time of 180 s. However, the turn-up time was reduced to 120 s in cases where ECG abnormalities were detected.”

(7) What is the quantitative assessment of how “uniform” the lesions are?

“The gradual temperature rise mode showed superior performance by creating larger and more uniform lesions compared to fixed temperature settings at 75°C and 85°C.”

Author Response

The article by Cheol-Min Lee et al. describes a set of comparisons between the gradual temperature rise mode vs temperature-controlled mode of Radiofrequency ablation (RFA) on in-vitro and in-vivo experiments with porcine myocardial tissue.

The reviewer greatly appreciates the efforts of the authors, especially in the quality of study design, experimental execution, writing and presentation. The reviewer has a RITA Radiofrequency Ablation System (AngioDynamics) in the lab and shares the frustration of controlling the ablative process. The temperature rise mode proposed by the authors has the potential to address some shortcomings of current RFA procedures.

Before endorsing for publications, there are a few concerns over the current version of the articles. To be specific:

(1) The implementation of “Gradual temperature rise in radiofrequency ablation”

Since this is a new control method, at least the reviewer hasn’t been aware of, more details of this new approach during RFA are needed. Even though some information, such as the diagram in Figure 2a, the overall descriptions are still oversimplified. Technical aspects, including but not limited to the following, are encouraged.

A : Thank you for your comment. The gradual temperature rise method in RFA is a new approach designed to optimize energy transfer and lesion formation while enhancing safety. This method gradually increases the temperature in controlled increments (e.g., from 65°C to 85°C in 5°C steps) rather than maintaining a fixed temperature, reducing the risk of overheating and improving lesion uniformity. We have provided additional details on the implementation process, including the RF generator control mechanism and procedural parameters, in the revised manuscript. We have also expanded the description in Figure 2a to provide a clearer illustration of the gradual temperature rise technique.

  • Temperature sensor

Where exactly is the temperature and what temperature does it really measure? Figure 2b looks like it’s measuring the temperature of the “cooling lumen” other than the tissue in contact.

A : Thank you for your feedback. The temperature sensor is positioned between the electrode and the cooling lumen to measure the temperature of the tissue that the electrode is in direct contact with. The cooling lumen helps to cool the electrode temperature to prevent carbonization, but the temperature sensor itself is positioned to provide accurate feedback on tissue heating during ablation. To clarify this, we have revised the description and improved Figure 2b to more accurately describe the location of the sensors that monitor tissue temperature as well as the cooling lumen.

  • control/adjustment logic

How does the voltage, current or power of the RFA being controlled with temperature inputs? Examples of the temperature history would be helpful, especially demonstrating how well the system achieves “Gradual temperature rise”

A : Thank you for your comment. The RFA system operates with an input voltage of 220V, and the power output is dynamically adjusted based on temperature feedback.  The output energy is in watts and ranges from 3W to 10W depending on the set temperature, tissue impedance, and whether or not a cooling system is present. The maximum output energy is limited to 20W. The energy output is initially high to achieve the target temperature, but once the target temperature is reached, minimal watts are used to maintain the temperature. Method 2.1 covers the generator, but we have included additional explanations in the manuscript.

  • temperature history during RFA

Examples of the temperature history would be helpful, especially demonstrating how well the system achieves “Gradual temperature rise”.

A : The temperature changes instantaneously because the RF generator controls the temperature by injecting the right amount of energy to reach it. We've added the graph below to the figure.

  • the turn-up time

Examples of the temperature history would help.

In the graph above, the turn up time in gradual temperature rise mode is 120 seconds.

(2) The actual device(s)

It looks to me that a few devices were made (in section 2.1. RFA System and RF Catheter) and used in the experiments with different geometry: an outer diameter of 4–5 Fr: 1.33mm (for 4 Fr) to 1.67mm (for 5 Fr) and electrode lengths of 10–15 mm.

A fair comparison should be made using the same device, otherwise the difference in lesion formation can be attributed to different sizes not the modes of operation (constant temp. vs gradual temperature rise). Please confirm your device geometries of all the treatments.

A : Thank you for your comment. We confirm that all treatments in the experiment were conducted using RF catheters of the same geometry to ensure a fair comparison. Specifically, we used a catheter with a consistent outer diameter of 5 Fr (1.67 mm) and an electrode length of 15 mm across all experimental groups. The mention of 4–5 Fr and 10–15 mm in Section 2.1 refers to the general specifications of the devices available, but for this study, only the 5 Fr, 15 mm configuration was used in both the gradual temperature rise and constant temperature modes. We have clarified this in the revised manuscript to prevent any misunderstanding.

(3) The overall advantages of the “gradual temperature rise mode” over “temperature-controlled mode” should be made clearer.

If one only considers size (the larger the better), the microwave ablation is the clear winner.

If the shape (the more spherical the better) is concerned, then a smaller probe of the RFA will be needed.

If the aim is to avoid carbonation (scar tissue) and vaporization (phase change), then lower temperature and longer RFA time would help.

In other words, what new capability does the “gradual temperature rise mode” offer that cannot be achieved by other means? What specific problems do the “gradual temperature rise mode” solve that cannot be solved by other designs?

A : The key to the gradual temperature rise mode is to form a stable and efficient lesion without carbonization within the same amount of time. While microwave ablation generates larger lesions, it is not suitable for applications requiring precise lesion formation, such as myocardial ablation. For interventricular septal ablation, an elongated oval lesion is more effective than a spherical one and smaller RFA probes can improve lesion shape but may reduce overall ablation efficiency.  Lower temperatures and longer RFA durations may not result in efficient lesion shaping due to increased tissue impedance.

Other minor questions or suggestions, in no particular order.

(1) The introduction has only 1 paragraph and is fairly long. Breaking it down to a few smaller ones is recommended.

A : Thank you for your suggestion. We agree that the introduction is relatively long and could be more reader-friendly if divided into smaller paragraphs. In the revised manuscript, we have restructured the introduction into multiple paragraphs, grouping related information logically to improve readability and clarity.

(2) More clear definitions of the lesion quality (lesion formation quality) and safety is greatly appreciated, especially in the context of hypertrophic obstructive cardiomyopathy (HOCM) Radiofrequency ablation (RFA).

A : Thank you for your valuable comment. We acknowledge the importance of clearly defining lesion quality and safety, particularly in the context of hypertrophic obstructive cardiomyopathy (HOCM) radiofrequency ablation (RFA). In the revised manuscript, we have provided a more precise definition of lesion formation quality, including parameters such as lesion size, uniformity, depth, and avoidance of undesirable effects like carbonization and steam pops. Additionally, we have clarified the safety aspects by discussing factors such as controlled energy delivery, prevention of excessive heating, and mitigation of complications like impedance spikes and collateral tissue damage. These refinements should better contextualize the clinical significance of the gradual temperature rise mode in HOCM RFA procedures.

(3) How is the lesion defined? From figure 3a, a transient zone exists between the pink outer area and the inner white area.

A : Thank you for your question. In this study, the lesion is defined based on the distinct thermal effects observed in the ablated tissue. Specifically, the lesion boundary is determined by the region of coagulative necrosis, which appears as the central white area in Figure 3a. The pink outer area represents a transitional zone where partial thermal effects occur, but this region is not included in the lesion measurement as it does not exhibit complete necrosis. To ensure consistency, lesion size is quantified based on the maximum diameter of the inner coagulative necrosis zone. We have clarified this definition in the revised manuscript.

(4) The first in vitro experiment, the reviewer does not understand why (ii) has smaller ablative size and lower energy than (iii). Wouldn’t maintaining the RFA probe at higher temperature (85C) require more power than that at a lower temperature (65-85)?

(i) a fixed temperature of 75°C for 10 min,

(ii) fixed temperature of 85°C for 10 min, and

(iii) gradual temperature rise mode, wherein the temperature was incrementally increased from 65°C to 85°C at a rate of 5°C every 2 min over a total duration of 10 min.

A : Thank you for your question, Figure 3c should help explain this. Our RF generator only injects the minimum amount of energy needed to maintain a stable temperature, so the energy injection may be higher during the process of increasing the temperature than when maintaining a constant temperature. Thus, an energy injection pattern of 65-85 degrees may be more effective than a fixed 85 degrees. We have explained this point more clearly in the revised manuscript.

(5) The 2nd in vitro experiment, the following description on “Turn-up time” is confusing. Again, plots of temperature histories would help.

“The second experimental setup determined the optimal turn-up time for a gradual temperature increase during ablation. Turn-up time refers to the time interval in which the temperature increases by 5°C. Internal catheter cooling was performed at a saline flow rate of 2 cc/min in all experimental groups. Ablation began at an initial temperature of 60°C, with temperature increments of 5°C every 2 min until 85°C was reached. Ablation was terminated when impedance exceeded 250 Ω, and five turn-up times of 30, 90, 120, 150, and 180 s were tested (n = 6 per condition).”

A : Thank you for your feedback. We recognize that our description of “turn-up time” in the second in vitro experiment may be confusing. We have defined turn-up time to mean the interval of time during which we increase the temperature. For example, if the turn-up time is 120 seconds, we increase the target temperature by 5°C every 120 seconds. We have also included a temperature history plot in the revised manuscript to better illustrate the gradual increase in temperature over time for clarity.

(6) The in vivo experiment, what exactly are each of the treatment conditions? There are variations of the thermal conditions, making comparisons somewhat unfair.

“In the fixed temperature mode, ablation was conducted for a pre-determined duration at either 75°C or 85°C. In the gradual temperature rise mode, the temperature was incrementally increased from 60°C to 85°C in 5°C increments.  The ablation and temperature varied from 75°C to 85°C. The gradual temperature rise mode involved a turn-up time of 180 s. However, the turn-up time was reduced to 120 s in cases where ECG abnormalities were detected.”

A : The implication of this result extends beyond lesion size; in vivo animal experiments involve many variables, such as impedance changes and ECG responses. Therefore, a gradual temperature increase starting at 60 degrees may be more effective than a fixed controlled mode for impedance management and energy injection.

(7) What is the quantitative assessment of how “uniform” the lesions are?

“The gradual temperature rise mode showed superior performance by creating larger and more uniform lesions compared to fixed temperature settings at 75°C and 85°C.”

A : Thank you for your comment. The uniformity of the lesions was assessed based on their shape, boundary definition, and consistency across different samples. Specifically, lesion uniformity was quantified by measuring the ratio of the maximum to minimum diameters of the ablated area, as well as by evaluating histological cross-sections to assess consistent thermal penetration and the presence of irregular boundaries. A lower ratio indicates a more spherical and uniform lesion, while irregular or elongated shapes suggest non-uniformity. In our analysis, the gradual temperature rise mode produced lesions with a more consistent shape and well-defined boundaries, whereas fixed temperature modes often resulted in more irregular lesion geometries due to abrupt heating. We have added this quantitative assessment to the revised manuscript for better clarity.

Reviewer 2 Report

Comments and Suggestions for Authors
  1. Clearly define the research objective and hypothesis at the end of the introduction to provide a structured rationale for the study.
  2. Explain why these sample sizes were chosen to ensure robust and generalizable conclusions.
  3. The study concludes that the gradual temperature rise mode enhances lesion formation and safety, but it lacks an in-depth discussion of the biophysical mechanisms that support this improvement.
  4. Include a discussion on how these results could influence clinical protocols and whether they align with current clinical guidelines.
  5. Explicitly mention limitations and suggest future studies that could validate these findings in human trials or larger sample sizes.

Author Response

(1) Clearly define the research objective and hypothesis at the end of the introduction to provide a structured rationale for the study.

A : Thank you for your comment. We agree that clearly defining the research objective and hypothesis at the end of the introduction will provide a more structured rationale for the study. In the revised manuscript, we have explicitly stated the research objective as evaluating the effectiveness and safety of the gradual temperature rise mode compared to fixed temperature ablation in RFA, particularly in the context of lesion formation and procedural safety. The hypothesis is that the gradual temperature rise mode improves lesion uniformity, reduces carbonization, and enhances energy penetration while maintaining procedural safety. This revision strengthens the logical flow of the introduction and better frames the study’s significance.

(2) Explain why these sample sizes were chosen to ensure robust and generalizable conclusions.

A : Thank you for your comment. The sample sizes were determined based on previous studies in similar RFA experiments to ensure statistical robustness and reproducibility while considering practical constraints. For the in vitro experiments, a sample size of six per group (n = 6) was selected to allow for sufficient statistical power in comparing lesion formation characteristics while maintaining feasibility in terms of experimental resources. For the in vivo experiments, a total of ten pigs were used, with a distribution that ensured meaningful comparisons between treatment conditions while minimizing ethical and logistical limitations. Power analysis was conducted to confirm that these sample sizes would provide adequate sensitivity to detect significant differences in lesion size, energy transfer, and safety parameters. We have clarified this rationale in the revised manuscript to reinforce the validity and generalizability of our findings.

(3) The study concludes that the gradual temperature rise mode enhances lesion formation and safety, but it lacks an in-depth discussion of the biophysical mechanisms that support this improvement.

A : Thank you for your insightful comment. We acknowledge the importance of providing a more in-depth discussion of the biophysical mechanisms underlying the advantages of the gradual temperature rise mode. In the revised manuscript, we have expanded on the biophysical principles that support improved lesion formation and safety. Specifically, we explain how gradual temperature increments allow for deeper energy penetration by reducing the rapid impedance rise that occurs with fixed high-temperature ablation. This prevents early tissue dehydration and carbonization, thereby optimizing energy transfer efficiency. Additionally, a controlled temperature increase minimizes thermal gradients within the tissue, reducing the risk of steam pops and excessive heat diffusion, which can lead to collateral tissue damage. The gradual temperature rise also ensures a more predictable and uniform lesion shape by allowing sufficient time for thermal conduction to reach deeper tissue layers. These biophysical explanations have been integrated into the discussion section to strengthen the rationale behind the observed improvements in lesion formation and procedural safety.

(4) Include a discussion on how these results could influence clinical protocols and whether they align with current clinical guidelines.

A : Thank you for your valuable comment. We acknowledge the importance of discussing the clinical implications of our findings and their alignment with current clinical guidelines. In the revised manuscript, we have expanded the discussion on how the gradual temperature rise mode could influence clinical protocols for radiofrequency ablation (RFA), particularly in the treatment of hypertrophic obstructive cardiomyopathy (HOCM) and other cardiac ablation procedures. The improved lesion uniformity, enhanced energy penetration, and reduced risk of carbonization and steam pops suggest that this technique may offer a safer and more effective alternative to conventional fixed-temperature ablation. Given that current clinical guidelines emphasize procedural safety and lesion predictability, integrating a gradual temperature rise approach could help reduce complications while maintaining effective ablation outcomes. Furthermore, as temperature-controlled RFA is already widely used in clinical practice, refining this method through gradual heating could improve treatment efficacy without requiring major modifications to existing protocols. We have incorporated this discussion into the revised manuscript to highlight the potential translational impact of our findings.

(5) Explicitly mention limitations and suggest future studies that could validate these findings in human trials or larger sample sizes.

A : Thank you for your comment. We recognize the importance of explicitly addressing the limitations of this study and suggesting future research directions. In the revised manuscript, we have outlined key limitations, including the use of a porcine model, which may not fully replicate human cardiac physiology, and the relatively small sample size, which, while sufficient for statistical analysis, may limit generalizability to broader clinical applications. Additionally, while the gradual temperature rise mode demonstrated improved lesion formation and safety, long-term outcomes and potential clinical benefits require further validation. To address these limitations, we suggest future studies involving larger animal models or clinical trials in human patients to confirm the efficacy and safety of this approach in real-world settings. Additionally, studies investigating long-term lesion durability, arrhythmia recurrence rates, and potential refinements in temperature control protocols would provide further insights. These points have been incorporated into the revised manuscript to provide a more comprehensive perspective on the study’s findings and their implications.

Round 2

Reviewer 1 Report

Comments and Suggestions for Authors

The reviewer appreciates the authors’ timely revision and addressing the comments. The added description greatly improves the clarity of this article. 
A few minor suggestions:
1. The last paragraph of the introduction. I would recommend explaining a little bit more of “This gradual heating strategy enhances lesion formation quality, reduces damage to the surrounding tissue, and improves energy efficiency.”, especially in the context of HOCM RFA. There are good arguments in the author’ response to report, the reviewer would recommend highlighting the advantages of the “gradual temperature rise mode” over “temperature-controlled mode” here. The current wordings of “enhances lesion formation quality, reduces damage” were somewhat too general and non-specific. 
2. Representative temperature histories showing the differences in the “Turn-up time” can be helpful. 
3. Figure 3, power and temperature plots of the first group (75C for 10min) can be helpful to include. 
4. “Power” vs “Energy”. 
Power (일률) should represent the instantaneous rate of energy output (dW/dt), W is work, and t is time, in the unit of watt (W); while the energy (에너지) should take the unit of Joule. The two words are somewhat misused throughout this article, including in the figures and tables. 

Author Response

The reviewer appreciates the authors’ timely revision and addressing the comments. The added description greatly improves the clarity of this article. 
A few minor suggestions:
1. The last paragraph of the introduction. I would recommend explaining a little bit more of “This gradual heating strategy enhances lesion formation quality, reduces damage to the surrounding tissue, and improves energy efficiency.”, especially in the context of HOCM RFA. There are good arguments in the author’ response to report, the reviewer would recommend highlighting the advantages of the “gradual temperature rise mode” over “temperature-controlled mode” here. The current wordings of “enhances lesion formation quality, reduces damage” were somewhat too general and non-specific.

A : Thank you for your feedback. We will provide a more detailed explanation in the last paragraph regarding how the "gradual temperature rise mode" offers advantages over the "temperature-controlled mode," particularly in the context of HOCM RFA. We will emphasize how this approach improves lesion formation quality, minimizes collateral tissue damage, and enhances energy efficiency. To address this, we will incorporate key arguments from our author response letter, ensuring that the revised section clearly articulates the benefits of gradual temperature rise. The updated paragraph will focus on lesion uniformity, changes in tissue elasticity, and the reduction of excessive thermal damage, making the explanation more specific and compelling.

2. Representative temperature histories showing the differences in the “Turn-up time” can be helpful.

A : Thank you for your valuable feedback. To better illustrate the differences in "Turn-up time," we will add a representative figure showing the corresponding temperature histories. This addition will help readers intuitively understand the variations in temperature rise rates. Thank you.

3. Figure 3, power and temperature plots of the first group (75C for 10min) can be helpful to include. 

A : Thank you for your valuable feedback. We will add the power and temperature plots for the first group (75°C for 10 minutes) to provide clearer data. This addition will help readers better understand the experimental results. Thank you.

4. “Power” vs “Energy”. 
Power (일률) should represent the instantaneous rate of energy output (dW/dt), W is work, and t is time, in the unit of watt (W); while the energy (에너지) should take the unit of Joule. The two words are somewhat misused throughout this article, including in the figures and tables.

A : As per the reviewer’s suggestion, we have revised the manuscript to differentiate instantaneous power output as "Power (W)" and total accumulated energy as "Energy (J)." To ensure clarity and consistency, we have carefully reviewed and corrected the usage of these terms throughout the text, figures, and tables.
